# Characterization of Tunneled Wide Band Gap Mixed Conductors: The Na_2_O-Ga_2_O_3_-TiO_2_ System

**DOI:** 10.3390/nano13142054

**Published:** 2023-07-12

**Authors:** Javier García-Fernández, María Hernando, Almudena Torres-Pardo, María Luisa López, María Teresa Fernández-Díaz, Qing Zhang, Osamu Terasaki, Julio Ramírez-Castellanos, José M. González-Calbet

**Affiliations:** 1Inorganic Chemistry Department, Chemical Sciences Faculty, Universidad Complutense de Madrid, 28040 Madrid, Spain; marher@ucm.es (M.H.); atorresp@ucm.es (A.T.-P.); marisal@ucm.es (M.L.L.); jgcalbet@quim.ucm.es (J.M.G.-C.); 2Institute Laue-Langevin, 71 Avenue des Martyrs, CS 20156, CEDEX 9, 38042 Grenoble, France; ferndiaz@ill.fr; 3Centre for High-Resolution Electron Microscopy, ShanghaiTech University, Shanghai 201210, China; zhangqing1@shanghaitech.edu.cn (Q.Z.); osamuterasaki@mac.com (O.T.); 4ICTS National Center for Electron Microscopy, Universidad Complutense de Madrid, 28040 Madrid, Spain

**Keywords:** nano-characterization, neutron powder diffraction, (Cs)-corrected electron microscopy, impedance spectroscopy, wide band gap semiconductor

## Abstract

This article focuses on the Na_2_O-Ga_2_O_3_-TiO_2_ system, which is barely explored in the study of transparent conductive oxides (TCOs). Na_x_Ga_4+x_Ti_n−4−x_O_2n−2_ (n = 5, 6, and 7 and x ≈ 0.7–0.8) materials were characterized using neutron powder diffraction and aberration-corrected scanning transmission electron microscopy. Activation energy, as a function of different structures depending on tunnel size, shows a significant improvement in Na^+^ ion conduction from hexagonal to octagonal tunnels. New insights into the relationship between the crystal structure and the transport properties of TCOs, which are crucial for the design and development of new optoelectronic devices, are provided.

## 1. Introduction

Transparent conductive oxides (TCOs) are a specific field of wide band gap semiconductors, which are traditionally recognized for their high conductivity and transparency in the visible range [1,2]. In recent years, significant efforts have been made to continually upgrade optoelectronic devices, largely dependent on the study and development of new TCOs [3,4,5,6]. One such example is Ga_2_O_3_, specifically its β-polymorph, which has gained attention due to its unique properties, including a wide band gap of 4.9 eV, a high electric breakdown field of 8 MV/cm and high thermal and chemical stability [7,8].

Several Ga-based materials have been studied for their remarkable applications, particularly in the development of transparent electrodes and optoelectronic devices. For example, the homologous series Ga_2_O_3_(ZnO)_m_ [9], as well as Ga_3−x_In_3_Ti_x_O_9+x/2_ [10], In-Ga-Zn-O [11], and Ga-Sn/Sb-Zn-O [12,13] systems, have been extensively researched. In all these systems, small variations in the composition of the cationic sublattice can cause significant changes in their electronic configuration and properties. In this article, we focus on the Na_2_O-Ga_2_O_3_-TiO_2_ system, which has barely been explored. Moreover, one advantage is the replacement of Ga, a critical element [14], by abundant and non-toxic elements such as Na and Ti.

Specifically, we investigate the n = 5, 6, and 7 members of the Na_x_Ga_4+x_Ti_n−4−x_O_2n−2_ (x ≈ 0.7–0.8) homologous series. Chandrasekhar et al. [15] synthesized and characterized the Na_0.7_Ga_4.7_Ti_0.29_O_8_ phase, which was later confirmed by Amoroso et al. [16] in their phase stability studies. Michue et al. described the synthesis and characterization by single-crystal X-ray diffraction (XRD) of the next two phases of the series, Na_0.8_Ga_4.8_Ti_1.2_O_10_ [17,18] and Na_0.8_Ga_4.8_Ti_2.2_O_12_ [19]. The structure of these oxides can be described as an ordered intergrowth of β-Ga_2_O_3_-type chains and Ga/Ti octahedra for the n = 5 (hereinafter referred to as NGT1O) and n = 6 (NGT2O) members, with a monoclinic *C2/m* space group. Because of this intergrowth, hexagonal tunnels are formed for the NGT1O compound and octagonal for NGT2O. However, the structure of n = 7 material (NGT3O) presents both edge-sharing and corner-sharing (Ga,Ti)O_6_ octahedra, as well as corner-sharing GaO_4_ tetrahedra with an orthorhombic *Pbam* space group. In all cases, Na^+^ ions are located in the tunnels of the structure. Our previous study [20] confirmed the structure of these materials by X-ray diffraction and aberration-corrected scanning transmission electron microscopy (STEM). Cathodoluminescence and photoluminescence response for the first two samples (NGT1O and 2) is essentially composed of a broad luminescence band centered around 2.2–2.3 eV, whereas the NGT3O sample’s luminescence is composed of an intense band centered at 1.6 eV [20]. Furthermore, the band gap values obtained for each phase were Eg1 = 4.2 eV, Eg2 = 4.1 eV, and Eg3 = 3.5 eV, showing a decrease in energy values as the Ti content increases [20]. In addition, because these materials contain Na^+^ situated in one-dimensional tunnels, a preliminary analysis of the potential of utilizing them as anode material has been explored using complex impedance. The characterization of materials with Na in their composition is a non-trivial task. Light and mobile elements made the use of conventional characterization techniques, such as XRD, more difficult. In this sense, techniques such as neutron diffraction and aberration-corrected transmission electron microscopy become essential techniques for the study of materials with light elements at the atomic level.

In this work, we report an extensive and detailed characterization of these materials using neutron diffraction, transmission electron microscopy, and complex impedance as a first step in the study of these materials as possible candidates in potential sodium ion anodes [21,22,23]. This, added to their interesting optical and photocatalytic and biomaterial properties reported before [24,25], makes these materials potential multifunctional oxides of technological interest in the field of optoelectronics.

## 2. Materials and Methods

Na_x_Ga_4+x_Ti_n−4−x_O_2n−2_ (n = 5, 6 and 7; x ≈ 0.7–0.8) powders were prepared by a solid-state reaction using the following starting reagents: Na_2_CO_3_ (99.5%, Aldrich, St. Louis, MO, USA), Ga_2_O_3_ (99.99%, Aldrich, St. Louis, MO, USA) and TiO_2_ (99.9%, Aldrich, St. Louis, MO, USA). The powders were pressed into pellets and heated up to 1225 °C inside an alumina crucible using sacrificial powders to avoid contamination and prevent sodium losses. The reaction times were 48 h for the so-called NGT1O (n = 5) and NGT3O (n = 7) and 14 h for NGT2O (n = 6). After that, the crucible was quenched in air. Morphological characterization by transmission electron microscopy (TEM) did not present any particular morphology.

Neutron powder diffraction (NPD) measurements were performed at the Institut Laue Langevin, Grenoble (France) on a D2B diffractometer (λ = 1.594 Å). [26]. The data were analyzed using the Rietveld method [27] using the software package FullProf [28]. The structural model of both phases was displayed using VESTA software [29].

Cationic quantification was performed using a JEOL Superprobe JXA-8900M (JEOL, Tokyo, Japan) microscope, equipped with five wavelength dispersive X-ray spectrometers (WDS), allowing analysis over a sample diameter of 5–10 μm. The voltage and current used in the analysis of the Na-Ga-Ti-O samples were 20 kV and 50 nA, respectively.

Atomic resolution transmission electron microscopy was performed on a JEOL JEM GRANDARM 300F (JEOL, Tokyo, Japan) equipped with cold FEG and double Cs correctors operating at 300 kV and a JEOL JEM-ARM 200cF (JEOL, Tokyo, Japan) equipped with a cold FEG and probe Cs-corrected operating at 200 kV. High-angle annular dark-field (HAADF) images were recorded using a nominal camera length of 100 mm and inner and outer collection semi angles of 64 and 180 mrad (JEOL JEM GRANDARM 300F) and a nominal camera length of 60 mm and inner and outer collection semi angles of 68 and 280 (JEOL JEM-ARM 200cF). For (S)TEM observations, the samples thus prepared were crushed in an agate mortar and ultrasonically dispersed in n-butanol and then transferred to carbon-coated copper grids. A structural characterization by X-ray diffraction of the samples investigated here was performed in our previous work [20].

The electrical measurements were performed by pressing powders of NGTO under 6 kbar, into pellets of 8 mm diameter and around 1 mm thickness, and sintering at 1225 °C. For the a.c. electrical conductivity measurements, blocking electrodes were deposited on both sides of the as-pelletized samples using platinum paint (previously dried at 500 °C) in the 573–753 K temperature range. The a.c. conductivity data were obtained using a frequency analyzer (Solartron 1260) coupled with a dielectric interface (Model 1296A) over a frequency range of 1 MHz–0.1 Hz and under oscillation voltages of 50 mV. Analysis of impedance spectra was performed using Zview 3.5f software (Scribner Associates) by fitting the data to appropriate equivalent circuits in order to resolve the impedance response into bulk, grain-boundary and electrode contributions.

## 3. Results and Discussion

In the first step, we assessed the precise structure of the Na-Ga-Ti oxides by neutron diffraction to gain a deeper insight into their structure and composition. Because the scattering lengths of Ga and Ti atoms are too distinct (bGa = 7.288 fm; bTi = −3.438 fm), they were solved by NPD and thus we obtained the Ga/Ti ratio in each polyhedron of the structure. Moreover, this technique allows determining of the oxygen and the Na content. The details of the analysis of the neutron diffraction data are described herein for NGT1O and NGT3O.

For NGT1O, the NPD pattern was refined using the monoclinic isostructural oxide, Na_0.7_Ga_4.7_Ti_0.29_O_8_ [15] with a *C2/m* space group as a starting structural model. Figure 1a shows the results of the final NPD data fitting, whereas the corresponding structural parameters and distances are included in Table 1 and Appendix A, respectively. The isotropic temperature factors were also refined, except for the Na^+^ cation, for which an anisotropic factor was refined. The refinement of occupancy factors of the oxygen atoms does not reveal anionic vacancies; therefore, these factors were fixed during the refinement. The structure, schematically represented in Figure 1b, can be described on the basis of an ordered intergrowth of β-Ga_2_O_3_ type chains (formed by octahedral and tetrahedral sites, colored in red) and (Ga/Ti)O_6_ octahedra (colored in yellow).

The tetrahedral and octahedral sites were forced to be fully occupied but no constraint was imposed on the Ga: Ti ratio. The refinement shows that all tetrahedra are exclusively occupied by Ga (Ga1) whereas the octahedra ((M)2,3; M = Ga/Ti)) reveal the presence of both Ga and Ti with higher Ga concentrations. The distribution of Ga and Ti among octahedra is summarized in Table 1. Furthermore, the refinement shows that Na^+^ ions occupy approximately 40% of the total 4a Wyckoff position. The easy mobility of Na^+^ in the tunnel is reflected in the anisotropic factor temperature (Table 1) indicating that the Na^+^ ions are delocalized on the tunnel, which could suggest its potential use as an ionic conductor. The cell parameters obtained from the NPD refinement are: a = 12.36138(8) Å, b = 3.00098(2) Å, c = 9.36302(4) Å, β = 122.1140(4)°. Additionally, according to NPD refinement, the composition obtained was Na_0.80(3)_Ga_4.66(3)_Ti_0.34(3)_O_8_ with the complete anionic sublattice.

In Na_0.80(3)_Ga_4.66(3)_Ti_0.34(3)_O_8_, the GaO_4_ tetrahedra are quite regular with four Ga-O distances (see Appendix A) very similar to those found in β-Ga_2_O_3_ (average distance = 1.839 Å). The corner-shared octahedra M3O_6_ is nearly regular but, on the contrary, the edge-shared octahedra M2O_6_ is quite distorted (see distances in Appendix A). However, the presence of Ga and Ti in the octahedral sites does not significantly modify the Ga–O distances in an octahedral environment compared to the β-Ga_2_O_3_ (average distance Ga-O = 1.993 Å).

The NPD study of the NGT3O was performed assuming, as an initial model, the orthorhombic isostructural oxide Na_1−x_Ti_2+x_Ga_5−x_O_12_ (x = 0.2) with *Pbam* space group. [19]. The result of the final NPD data fitted is depicted in Figure 2a, whereas the corresponding structural parameters and distances are summarized in Table 2 and Appendix A, respectively. The crystal structure, depicted in Figure 2b, can be described by both edge-sharing and corner-sharing MO_6_ octahedra, as well as corner-sharing GaO_4_ tetrahedra. As can be observed, large channels are extended parallel to the c-axis in which Na+ ions are delocalized occupying 35% of the 4e positions. The chemical composition, obtained after refinement of the Ga and Ti occupancies in each polyhedron, corresponds to Na_0.70(1)_Ga_4.72(1)_Ti_2.28(1)_O_12_ with the anionic sublattice complete. The cell parameters obtained from the NPD refinement for this member of the series are: a = 15.8046(12) Å; b = 9.33302(7) Å; c = 2.998228(18) Å.

Again, the refinement shows that all tetrahedra are exclusively occupied by Ga (Ga1) whereas the octahedra reveal the presence of Ti and Ga (see Table 2). The M4 octahedron is quite regular and is occupied preferably by Ga (85%), colored yellow in Figure 2b, as previously found in Na_0.80_Ga_4.66_Ti_0.34_O_8_ (NGT1O). However, M2,3 octahedra are quite distorted (see distances in Appendix A) and a distinct Ga:Ti ratio is present, in contrast to those observed in Na_0.80_Ga_4.66_Ti_0.34_O_8_ that have the same Ga:Ti concentration. Whereas the M2 site is richer in Ga (83%), the M3 site is occupied mainly by Ti (89%), colored green in Figure 2b. This distribution of Ga and Ti in the different polyhedra of the framework, obtained from the refinement of the neutron diffraction pattern as well as all average distances (M-O), is in good agreement with that proposed for Na_1−x_Ti_2+x_Ga_5−x_O_12_ (x = 0.2) [19].

Unfortunately, the NPD pattern of NGT2O prepared for the neutron diffraction experiment shows a very weak presence of NGT1O and NaTi_2_Ga_5_O_12_ [30] as impurities. The LeBail-type refinement, shown in Appendix A, allowed us to obtain the lattice parameters a = 12.091(1) Å, b = 3.013(2) Å, c = 10.385(2) Å and β = 92.199(3)° in space group *C2/m*.

The overall sodium, gallium and titanium content was also analyzed by WDS, (Appendix A). The Ga:Ti ratio obtained for the NGT1O and NGT3O phases is in good agreement with that obtained by NPD refinement. However, a slight difference can be observed in the Na^+^ content due to the possible correlation between Na^+^ occupancy and the temperature factors because of its mobility in the tunnel. This mobility of Na^+^ indicates that this material could be a potential candidate for use in Na^+^ ion anodes. The activation energy of Na^+^ will be explored further by complex impedance spectroscopy.

The close structural features between the phases of this series require an in-depth and detailed structural study using transmission electron microscopy (TEM). As these materials can be used as potential ionic conductors, a more detailed analysis along the directions where the tunnels of the structure are observed has been carried out. Figure 3a,b shows representative STEM-HAADF images of a characteristic crystal of the NGT1O phase along the [010] zone axis and NGT3O along [001], respectively. In the STEM-HAADF image corresponding to NGT1O (Figure 3a), β-Ga_2_O_3_ chains (represented by red circles in the image) are clearly seen, intergrowing in an ordered way with GaO_6_ or TiO_6_ columns (marked with dark yellow circles), giving rise to an irregular hexagonal tunnel of the dimensions 4.5 × 3.5 Å. Despite the difference in the Z values of each cation (Z_Ga_ = 31 and Z_Ti_ = 22), any difference in contrast is practically indistinguishable, due to the multiple occupation of cations at this position and the low concentration of Ti (see the refinement by NPD for this oxide). A similar characterization has been done for NGT3O (Figure 3b). Appendix A shows a STEM-HAADF image corresponding to NGT2O along the [010] zone axis. In all the images, the cell parameters and the tunnel sizes, where the Na^+^ is located, are indicated. Moreover, all the observed crystals are apparently ordered, so the presence of extensive defects can be ruled out, as well as disordered intergrowths with other members of the series. A more detailed characterization of the NGT2O and NGT3O phases by STEM-HAADF and ABF together with EELS spectroscopy was performed in our previous work [20].

Because these materials present one-dimensional tunnels where Na^+^ is placed, a characterization by impedance spectroscopy has been carried out to study their transport properties. As only NGT1O and NGT3O samples are single phases we have focused the analysis on these two members. Onoda et al. [31], in their study by NMR of the ^23^Na nuclei, observed an increase in Na^+^ mobility as a function of temperature, suggesting its possible use as a candidate for Na^+^ ion batteries. Figure 4a–f shows, as an example, the complex impedance diagrams at three different and representative temperatures for NGT1O and NGT3O compounds. In sample NGT1O, there are two overlapping semicircles and a small tail tilted on the *x*-axis. For sample NGT3O, a curve with a well-defined semicircle followed by a semicircle that does not end up closing is obtained. Along with the impedance measurements, the equivalent circuits used to fit the data are also shown. In addition, the data obtained from the simulations for the indicated fits are plotted in Figure 4a–f. It is worth mentioning that, over the entire working temperature range (500–753 K), the experimental data can be adjusted to a circuit, as shown in Figure 4a,d, for the two measured compounds. The capacitance values of all the samples obtained are of the order of ~10^−11^ F in the case of the first semicircle, which makes it possible to associate them with a phenomenon taking place in the bulk of the material. At middle frequencies, 10^–11^–10^−8^ F·cm^−1^ values are obtained which are interpreted as being due to grain boundaries and, at the lowest frequencies, 10^−7^–10^−5^ F·cm^−1^ values are typical of sample-electrode interface effects. The values of capacitance and resistivity for all temperatures measured are included in Appendix A.

From the resistance values obtained from the simulation, the values of the bulk conductivity were calculated. Fitting to an Arrhenius-type behavior, log(σT) as a function of 1/T, the activation energies (Ea) of the conduction processes can be calculated. The experimental data are plotted in Figure 5, together with the line fittings.

The Ea values obtained were 1.10 and 0.88 eV for the NGT1O and NGT3O oxides, respectively. As can be seen, there is a significant decrease in the activation energy as the series member increases, indicating that the change from a hexagonal to an octagonal tunnel is especially important. A second factor to be considered would be the amount of sodium per unit formula, being slightly lower in the case of NGT3O, perhaps allowing more mobility along the tunnel in a hopping-type mechanism. The result for the NGT1O sample is higher than that obtained by Hasegawa et al. (0.39–0.57 eV) [32]. However, it should be pointed out that in their work, they have managed to synthesize the sample in an oriented manner along the tunnel direction, which would justify this lower value of activation energy. The values obtained for these compounds are characteristic of one-dimensional Na^+^ conductors and close to the ones reported in the literature. For example, values between 0.73 to 1.0 eV can be found for Na_2_Co_8_(PO_4_)_6_, Na_8_M_4_(P_2_O_7_)_4_ (M = Mn, Co, Ni) or NaGe_2_(PO_4_)_3_ [33,34,35].

## 4. Conclusions

In the present work, three members of the homologous series Na_x_Ga_4+x_Ti_n−4−x_O_2n−2_ (n = 5, 6 and 7 with x ≈ 0.7–0.8) have been synthesized. According to the study performed by powder neutron diffraction, it can be concluded that both NGT1O and NGT2O samples crystallize in the *C2/m* space group as intergrowths of β-Ga_2_O_3_ chains and Ga/Ti octahedra. The NGT3O sample crystallized in the *Pbam* space group and its structure can be described as being formed by both edge-sharing and corner-sharing (Ga,Ti)O_6_ octahedra, as well as corner-sharing GaO_4_ tetrahedra. In all cases, the structure contains tunnels where the Na^+^ cation is placed. This information, combined with aberration-corrected scanning transmission electron microscopy, has allowed us to obtain a global and atomic-scale structural confirmation of these materials. Critical and precise information about atomic coordinates and the chemical composition and occupancy of each atomic position together with a direct visualization of the tunnel size for the NGT1O and NGT3O oxides were obtained. This information is crucial to the study of the transport properties of these materials. The activation energies corresponding to each sample obtained by using impedance spectroscopy were: 1.10 and 0.88 eV for NGT1O and NGT3O, respectively. These values demonstrate a clear structure–properties relationship, with tunnel size being a key factor in Na^+^ ion transport, making these materials functional oxides of technological interest.

## Figures and Tables

**Figure 1 nanomaterials-13-02054-f001:**
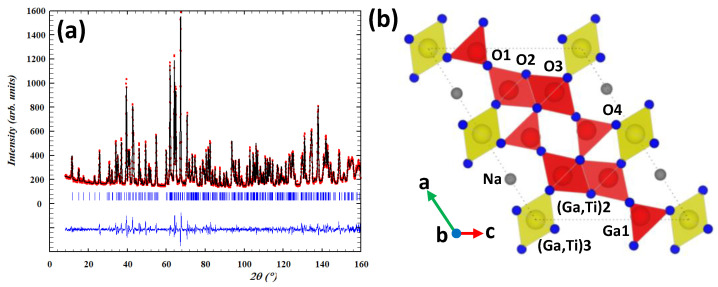
(**a**) Rietveld refinement of the NPD data for Na_0.80(3)_Ga_4.66(3)_Ti_0.34(3)_O_8_. The observed patterns (red circles), calculated patterns (continuous black line), and difference curves (continuous blue line) are shown. (**b**) Structure model for NGT1O oxide. Color code: red: β-Ga_2_O_3_ type chains, yellow: Ga/Ti octahedra, grey: (Na), blue: (O).

**Figure 2 nanomaterials-13-02054-f002:**
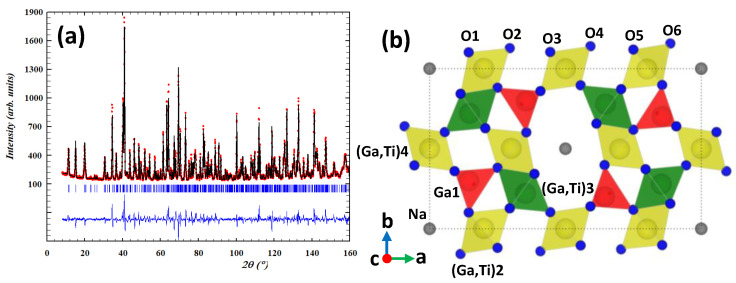
(**a**) Rietveld refinement of the NPD data for Na_0.70(1)_Ga_4.72(1)_Ti_2.28(1)_O_12_. The observed patterns (red circles), calculated patterns (continuous black line), and difference curves (continuous blue line) are shown. (**b**) Structure model for NGT3O oxide. Color code: red: Ga tetrahedra, yellow: M2 and M4 Ga rich-octahedra, green: M3 Ti rich-octahedra, grey: (Na), blue (O).

**Figure 3 nanomaterials-13-02054-f003:**
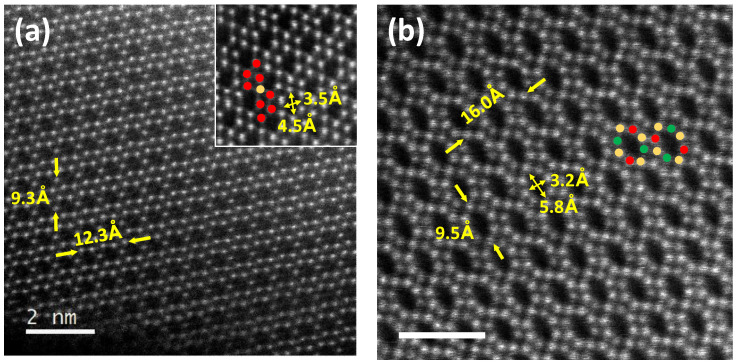
(**a**) HAADF-STEM image corresponding to (**a**) NGT1O along [010] and (**b**) NGT3O along [001]. Cell parameters and size of the hexagonal or octagonal tunnel (depending on the phase) are indicated in each image. Color code: Ga atomic columns in red, Ga/Ti atomic columns in dark yellow (Ga rich) or green (Ti rich). The scale bar is 2 nm.

**Figure 4 nanomaterials-13-02054-f004:**
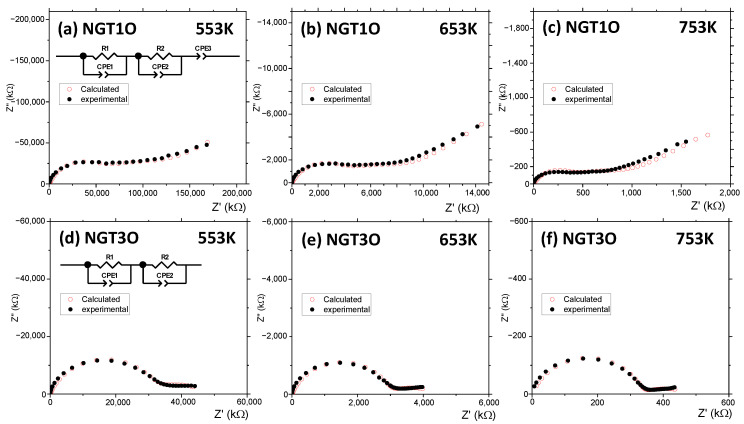
Impedance measurements for NGT1O (**a**–**c**) and NGT3O (**d**–**f**) at 553, 653, and 753 K.

**Figure 5 nanomaterials-13-02054-f005:**
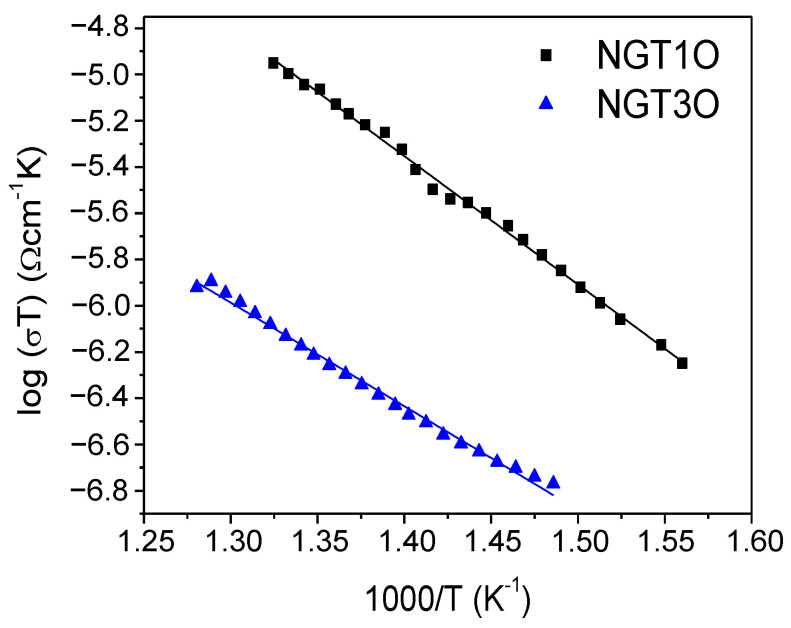
Evolution of the log(σT) as a function of temperature for NGT1O and NGT3O oxides.

**Table 1 nanomaterials-13-02054-t001:** Structural parameters from the refinement of neutron powder diffraction data for Na_0.80(3)_Ga_4.66(3)_Ti_0.34(3)_O_8_.

Atom	Wyckoff	x	y	z	Biso (Å^2^)	Occ.
Ga1	4i	0.4794(2)	0	0.3030(3)	0.49(3)	1
(Ga/Ti)2	4i	0.2432(2)	0	0.6443(3)	0.29(4)	0.930(4)/0.070(4)
(Ga/Ti)3	2a	0	0	0	0.68(8)	0.800(6)/0.200(6)
Na	4i	0.2368(14)	0	0.006(2)	*	0.400(14)
O1	4i	0.1027(3)	0	0.6847(3)	0.61(4)	1
O2	4i	0.3501(3)	0	0.5358(3)	0.38(4)	1
O3	4i	0.1721(3)	0	0.2134(3)	0.64(4)	1
O4	4i	0.4451(3)	0	0.0864(3)	0.63(4)	1

* Anisotropic parameter for Na: β_11_ = 0.0028(15), β_22_ = 0.21(3) β_33_ = 0.012(3). Fit parameters: R_B_ = 3.84, χ_2_ = 5.31.

**Table 2 nanomaterials-13-02054-t002:** Structural parameters from the refinement of neutron powder diffraction data for Na_0.70(1)_Ga_4.72(1)_Ti_2.28(1)_O_12_.

Atom	Wyckoff	x	y	z	Biso (Å^2^)	Occ.
Ga1	4g	0.1447(2)	0.3000 (4)	0	0.30(1)	1
(Ga/Ti)2	4i	0.1985(3)	0.9979(5)	0.5	0.44(1)	0.83(6)/0.17(6)
(Ga/Ti)3	4g	0.3545(8)	0.2261(12)	0	0.98(20)	0.11(1)/0.89(1)
(Ga/Ti)4	2d	0	0.5	0.5	0.47(1)	0.85(1)/0.15(1)
Na	4e	0	0	0.108(5)	*	0.35(3)
O1	4g	0.1439(3)	0.1004(4)	0	0.48(7)	1
O2	4h	0.2945(3)	0.1283(5)	0.5	0.31(6)	1
O3	4g	0.4445(3)	0.1011(4)	0	0.35(7)	1
O4	4h	0.0872(3)	0.3534(5)	0.5	0.41(6)	1
O5	4g	0.2529(3)	0.3784(4)	0	0.19(6)	1
O6	4h	0.3905(2)	0.3485(4)	0.5	0.34(6)	1

* Anisotropic parameter for Na: β_11_ = 0.0004(9), β_22_ = 0.040(8) β_33_ = −0.03(4). Fit parameters: R_B_ = 7.78, χ_2_ = 10.2.

## Data Availability

The data are available on reasonable request from the corresponding author.

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
