# Peer review of "Characterization of Tunneled Wide Band Gap Mixed Conductors: The Na2O-Ga2O3-TiO2 System"

_nanomaterials, 2023, doi:10.3390/nano13142054_

Round 1

Reviewer 1 Report

The authors report on the NPD, TEM and EIS characterization of NaGaTiOx samples, which they consider potential materials for Na-ion battery applications. Overall, the results do not represent a significant advance in the field and the conclusions derived do not bring much new information to the community (also considering ref. [20]). I leave it to the editor to decide if this matches what the journal wants.

Specific comments:

There are several simple and grammatical errors.

The partial electronic and ionic conductivities should be determined experimentally.

The potential usage as ion conductor and/or active material in Na-ion batteries is mentioned multiple times but no such data is shown. This is very confusing as the latter would warrant publication in this journal. I suggest the authors collect preliminary battery data, which would clearly strengthen the manuscript.

-

Author Response

The authors report on the NPD, TEM and EIS characterization of NaGaTiOx samples, which they consider potential materials for Na-ion battery applications. Overall, the results do not represent a significant advance in the field and the conclusions derived do not bring much new information to the community (also considering ref. [20]). I leave it to the editor to decide if this matches what the journal wants.

There are several simple and grammatical errors.

We thank the referee. Grammatical errors have been corrected.

 The partial electronic and ionic conductivities should be determined experimentally. The potential usage as ion conductor and/or active material in Na-ion batteries is mentioned multiple times but no such data is shown. This is very confusing as the latter would warrant publication in this journal. I suggest the authors collect preliminary battery data, which would clearly strengthen the manuscript."

We appreciate the referee's comments. However, we would like to highlight that this work is mainly focused on the detailed structural characterisation of Na materials that can be used as potential battery cathodes. With this work, we have carried out a study using techniques sensitive to Na content and a structural study at the atomic level, which is essential for its future applicability. A more in-depth study of the electrical and transport properties is needed, as the referee suggests, for future use of this material in potential devices. Nevertheless, we consider that it is outside the scope of this work. In consequence, we have modified the introduction and discussion (marked in yellow) of the manuscript as well as the title to reinforce our purpose of exploring and studying the structure of the material.

Reviewer 2 Report

This manuscript described the structure of Na2O-Ga2O3-TiO2 system as potential candidates for sodium ion batteries. This material structure is interested for the readers of this manuscript. However, the reviewer thinks that there is not enough discussion about why it is a promising material for sodium ion batteries.

In which part of the battery materials is the material you are discussing available? Cathode material? Anode material? Solid electrolyte?

If so, which metals can contribute to redox? Titanium contains only about 0.4 mol. The gravimetric energy density of this material looks very small.

In addition, only the activation energy is discussed in the manuscript, if the use as a solid electrolyte is considered. I think you should do a Na ion conductivity measurement for this material.

Author Response

This manuscript described the structure of Na2O-Ga2O3-TiO2 system as potential candidates for sodium ion batteries. This material structure is interested for the readers of this manuscript. However, the reviewer thinks that there is not enough discussion about why it is a promising material for sodium ion batteries.

In which part of the battery materials is the material you are discussing available? Cathode material? Anode material? Solid electrolyte? If so, which metals can contribute to redox? Titanium contains only about 0.4 mol. The gravimetric energy density of this material looks very small.

This article focuses on the Na2O-Ga2O3 -TiO2 system, the chemical formula of the phases can be described as NaxGa4+xTin-4-xO2n-2 (n = 5, 6, and 7 and x ≈ 0.7-0.8). The first member Na0.80(3)Ga 4.66(3)Ti0.34(3)O8 (NGT1O), due to the low content of Ti4+ only could be use as electrolyte. However, the conductivity obtained in the impedance experiment is too high, as it is seen in Nyquist plot (figure 4a) the grain boundary is very important, and to obtained the ionic conductivity real is necessary to reduce this effect and new synthesis are in progress. In contrary, the last compound of this series, Na0.70(1)Ga4.72(1)Ti2.28(1)O12 (NGT3O), could be used as anode in sodium ion batteries. With a theoretical specific capacity value of 95 mAh/g for this compound, taking into account that 2 Na+ ion could be inserted in the structure.

In addition, only the activation energy is discussed in the manuscript, if the use as a solid electrolyte is considered. I think you should do a Na ion conductivity measurement for this material.

We appreciate the referee's comments. However, we would like to highlight that this work is mainly focused on the detailed structural characterisation of Na materials that can be used as potential battery cathodes. With this work, we have carried out a study using techniques sensitive to Na content and a structural study at the atomic level, which is essential for its future applicability. A more in-depth study of the electrical and transport properties is needed, as the referee suggests, for future use of this material in potential devices. Nevertheless, we consider that it is outside the scope of this work. In consequence, we have modified the introduction and discussion (marked in yellow) of the manuscript as well as the title to reinforce our purpose of exploring and studying the structure of the material.

Round 2

Reviewer 1 Report

I am not satisfied with the (minor) changes made; the authors did not provide any new information. Besides, I am still not convinced that the results do represent an advance in the field (also considering ref. 20). As mentioned previously, the manuscript would strongly benefit from battery/conductivity data (see also comments from the other reviewer).

-
